# Exploring Factors Affecting the Acceptance of Genetically Edited Food Among Youth in Japan

**DOI:** 10.3390/ijerph17082935

**Published:** 2020-04-23

**Authors:** Mohamed Farid, Jianfei Cao, Yeongjoo Lim, Teruyo Arato, Kota Kodama

**Affiliations:** 1Graduate School of Technology Management—MOT, Ritsumeikan University, Osaka 567-8570, Japan; gr0347se@ed.ritsumei.ac.jp (M.F.); gr0401fv@ed.ritsumei.ac.jp (J.C.); 2Faculty of Business Administration, Ritsumeikan University, Osaka 567-8570, Japan; lim40@fc.ritsumei.ac.jp; 3Hokkaido University Hospital, Hokkaido University, Sapporo 060-8648, Japan; arato@med.hokudai.ac.jp

**Keywords:** genetically edited food, consumer acceptance, structural equation modeling, Japan, benefits, risk, willingness to purchase, science communication, regulations

## Abstract

Genetically edited food utilizes new techniques that may decrease all of the risks associated with genetically modified food, or “GMO” food. Safety and labeling regulations for genetically edited food are still new, and it is challenging for the consumer to differentiate it from conventional food. Although genetically edited food has the potential for reducing the risks associated with the gene introduction process, consumer perceptions toward it are still unclear. The research has compared the regulations governing GMO food and genetically edited food in Japan, Europe, and the United States. We found that the genetically edited food regulations in Japan are the most science-based, in the meaning that genetically edited food products are allowed to be sold without any safety evaluation. Based on the difference among regions, we further studied the potential acceptance level for such products among Japanese consumers, where regulation seemed science-based as policy. To understand the factors that may affect the adoption of genetically edited food among youth in Japan, we utilized the structural equation modeling (SEM) method with 180 surveys of Japanese university students to measure six factors: Knowledge, Attitude Towards Technology, Perceived Benefits, Perceived Risks, Trust, and Willingness to Purchase. The survey was conducted twice with an intervention in the middle to measure the effect of science communication, and we found significant differences when comparing the two datasets. The results of this survey indicate the importance of increasing knowledge and the positive role of science communication in increasing the adoption and trust of biotechnology products, such as genetically edited food.

## 1. Introduction

The world has witnessed a massive population growth, especially in the past 70 years. The world population has grown from 2.5 billion inhabitants in the year 1950 to nearly 7.7 billion in the year 2019, which means that the world population has tripled in less than a century [1]. Studies show that the world population will continue to grow to reach nearly ten billion inhabitants by the year 2050 [2]. One of the main challenges that the world faces now is food security. Therefore, the United Nations has created the Sustainable Development Goals (SDGs), making “No Hunger” the second of 17 goals to emphasize its importance [3]. The expectations for population growth in the upcoming decades show that more than half of the global population growth until 2050 will mainly take place in Africa [4]. The challenge intensifies given that Sub-Saharan Africa is facing a high level of food insecurity; studies estimate that nearly 220 million people were undernourished in 2016, which is a critical indicator for the future with the expected massive population growth [5]. The agriculture challenges are not just in developing countries but also in well-developed ones. With the aging population phenomenon in Japan and the significant change in social culture, the number of farmers dropped from 11 million in 1965 to less than 2 million in 2018. Food self-sufficiency in Japan is declining rapidly, as well. The domestic food production in Japan has dropped from over 70% in 1965 to less than 40% in 2018 [6]. Scientists around the world seek to solve these issues using food-engineering techniques, which can allow farmers around the world to produce food with high nutritional value, low cost, and less labor. The most popular technique in this field is utilizing genetically modified organisms (GMOs). GMOs have been adopted in 70 countries around the world [7], especially in the United States, where nearly 90% of soybeans and corn are produced using this method.

Genetically modified crops have been cultivated in the United States on a large scale since 1996, and the GMO crops cultivation area has exceeded 70 million hectares in 2015, putting the United States on the top of the countries that cultivate the genetically modified crops [8]. Nevertheless, there is no clear federal law or definition that clearly defines and regulates the GMO production process. However, the United States administration has created a framework that combines the efforts of three national entities to handle GMO related matters. The American governmental agencies in charge are; the U.S. Department of Agriculture (USDA), U.S. Environmental Protection Agency (EPA), and the U.S. Food and Drug Administration (FDA) [9]. On the other hand, the European Union has clear legislation and definitions for genetically modified organisms (GMO) as per the EU directive 2015/412. The EU has defined GMO as “an organism, with the exception of human beings, in which the genetic material has been altered in a way that does not occur naturally by mating and/or natural recombination” [10].

Although the GMO method is considered safe by scientific communities, consumers tend to have some concerns about it. Scientists have thus developed another methodology, which is a gene editing method that allows them to genetically edit crops without the use of any transgenes in a very accurate way. This process is considered scientifically safer, more cost-efficient, and quicker than the GMO method. However, there is a lack of knowledge about gene editing technology in food among average consumers as well as a lack of research about how consumers will perceive this type of new gene editing technology in food, especially that the regulatory process for it, is still ongoing [11].

This research study highlights the differences between the gene modification and gene editing technologies in food in terms of characteristics and regulations that govern them, especially in the markets of Japan, Europe, and the United States. The research also aims to investigate how the regulations affect innovation in this field. Based on the science-based regulation policy in Japan, the level of understanding and acceptance of gene editing technology for food among youth living in Japan was studied. In the study, we sought to uncover the general perception toward gene editing technology, and we also tried to identify the main factors that affect the acceptance of gene-edited food products. The research also intended to discover the role of knowledge and science communication in changing consumer perceptions toward gene editing food and highlight the importance of science communication in raising awareness for similar cases.

### 1.1. Genetically Modified Food

As mentioned in the introduction, there is no unified definition for genetically modified food. However, there are several definitions that define GMOs, such as the definition crafted by the World Health Organization (WHO)—it has defined GMO food as the food produced from or using genetically modified organisms. The WHO has also defined GMOs as organisms in which genetic materials (DNA) have been altered in a way that does not occur naturally [12]. The U.S. Department of Agriculture (USDA) defines the GMO as “An organism produced through genetic modification” [13]. The EU definition is aligned with the WHO definition.

In general, genetically modified food is food that contains genetically modified organisms (GMOs). In other words, the DNA of the organisms to be used as food has been altered or modified using biotechnology or without the natural recombination methodologies. This methodology is also often referred to as "DNA recombination" or "genetic engineering." Genetic modification technology allows for transferring a selected gene from one organism to another or between species that are not related.

The main reason behind developing GMO food is to provide food with a higher level of nutrition as well as a higher resistance to bacteria and herbicides. One of the most famous examples of GMO food is *Bacillus thuringiensis* corn or Bt-corn. It has been modified to produce specific proteins, which can act against insects or pests. Due to its unique characteristics, Bt-corn requires fewer pesticides, and it has a high level of herbicide resistance. Bt-corn is now widely used in the U.S. and has an 83% adoption rate [14].

The use of GMOs in food has resulted in a high level of controversy. Thus far, a majority of the scientific community has agreed that there is no proven evidence of significant risks to humans from consuming GMO food. A study published by “Critical Reviews in Biotechnology” in 2014 has shown that the researchers have reviewed all of the safety reports and scientific papers published regarding GMO food safety from 2002 to 2011; there is a scientific consensus based on this study that consumption of genetically engineered or genetically modified food is common worldwide and there is no sign of a hazard associated with this consumption [15]. Moreover, a major review conducted by The National Academies of Sciences, Engineering, and Medicine in 2016 discussed genetically engineered crops, and the committee concluded that there are solid scientific foundations and evidences that support the safety of genetically engineered crops [16]. Another study in the area of sustainable agriculture has identified a broader agreement—that no negative effect on health or environment has emerged because of commercializing genetically modified food. This was determined after fourteen years of continuously cultivating GMO crops and planting over two billion acres cumulatively. The study also notes that the research lab related to the European Commission has concluded that there is no substantial difference in the impact on human health between genetically modified crops and conventionally bred crops [17].

### 1.2. Genetically Edited Food

Genetic editing is considered one of the relatively new technologies for genome engineering. It is fundamentally different from genetic modification, in which gene editing precisely edits genes in a particular genome. The gene modification method usually inserts a foreign gene into the genome randomly in order to change the characteristics of the organism. However, gene editing inserts the edited gene into specific locations in the genome, which controls the outcome of the overall process. Different types of genome editors have been used in the past few years, but CRISPR-Cas9 is considered the most accurate technology. Currently, there are extensive research efforts toward utilizing gene editing technology, namely CRISPR, for food and agriculture. It has the potential to create very high value for farmers, as well as consumers, by eliminating any risk that might be associated with GMO food.

CRISPR is an acronym for Clustered Regularly Interspaced Short Palindromic Repeats. Scientists utilize this technology as molecular scissors to target a specific gene in the genome to be edited with a very high level of accuracy. The operation starts when the scientists recognize a specific gene in the genome that is responsible for a certain function in need of editing. Scientists create a guide RNA (short for ribonucleic acid), which simulates the DNA sequence that needs to be edited, as well as the Cas9 enzyme, which cuts the specific sequence at a targeted location. Specific functions can be added or edited after the cut, and after the change, the cell can be repaired using its enzymes. After the guide RNA and Cas9 enzyme are removed, the DNA editing process is considered complete, and the changes are implemented, similar to traditional breeding processes in the organism.

Since genome editing technology is considered scientifically safer than genome modification technology, regulatory authorities in Japan consider the food created by genome editing technology to be similar to the food created by conventional breeding. However, an online survey study conducted by the University of Tokyo, involving over 38,000 participants, showed that 43% of respondents expressed unwillingness to consume agricultural products created by genome editing technology. Only 9.3% of the respondents expressed a clear willingness to purchase and consume genetically edited food.

## 2. Literature Review

The implementation of novel technologies, such as bioengineering in the food industry, has continued to increase food industry productivity, reduce food prices, as well as enhance the nutritional values of food [18]. To identify the acceptance gap for genetically engineered food products, we have reviewed several papers in this regard. A 2015 review published by the journal *Viruses* argued that there is a large gap between farmers and consumers regarding the acceptance of genetically modified food [19]. The article illustrates a limited acceptance by consumers, citing factors such as consumers’ attitudes, knowledge, and perceptions about GMO food risks and benefits. To further understand the reasons behind GMO acceptance by farmers, another study analyzed nearly 147 studies in the field of agronomics, using mean impacts and meta-regressions to understand the performance of GMO crops around the world and identify the factors that affect these outcomes. The study showed that, on average, farmers experienced a 68% increase in profits due to the use of biotechnologies in agriculture. In addition, because of GMO crops, farmers have been able to reduce the use of chemical pesticides by 37%, which has led to an increase in crop yield by 22%, which shows the potential benefits for GMO food products [20].

The food supplies around the world today are massively affected by the technology in this field. However, based on several studies that looked at the diffusion of technology in the food industry, it was found that new technologies are often perceived with uncertainty and anxiety concerning the level of safety standards used [21]. Several previous studies have shown that consumers are often concerned about the potential negative effects that may result in consuming genetically modified food. The consumers have shown stronger concerns regarding the long-term negative potential effects such as; major health hazards, negative environmental effects, and potential effects on future generations. On the contrary, other consumers have expressed the potential positive effects of using genetically modified food, such as food price reduction, increase of a product’s shelf life, as well as reducing agricultural waste [22,23,24]. However, on a large scale, and in general, the concept of genetically modified food is often associated with negative impressions among the general public. A study by the European Council has shown that only 27% of the respondents tended to support genetically modified food in 2005. The percentage of supporters even decreased in the same study, during 2010, to only 23% of the respondents [25]. The cited studies have opened the door for more researchers to create more focused studies to identify factors that affect the acceptance of genetically modified food in different regions. A study conducted by Nara Medical University in Japan showed a public tendency to resist genetically edited food products in Japan, United States, United Kingdom, and France. Based on the opinions of the 1705 participants in the study, the highest level of resistance to GMO food was found in France, followed by Japan. Moreover, the study has shown that people without a high level of education have the highest level of resistance to GMO food, which shows the importance of knowledge and education in increasing the adoption of genetically engineered food products [26]. Another large-scale study of 4000 respondents was conducted in a high school in Belgium, to identify the variables that affect the willingness to eat GMO food among students. The study results have shown strong correlations between the willingness to eat GMO food and the subjective knowledge as well as the objective knowledge. The results of the study have confirmed the need for supporting the knowledge base of the students with more education about biotechnology and GMO related topics in the early-stage [27]. Another study focused on millennial attitudes towards genetically modified food in the United States; it showed that respondents with a high level of education and knowledge about biotechnology had a higher level of willingness to purchase genetically modified food products. Moreover, they tended to have a high level of perceived benefits and a lower level of perceived risks of using genetically edited food products [28]. Thus, the study has shown the importance of knowledge and education in increasing the adoption of genetically engineered food products.

One of the main sources of communication that plays an essential role in providing information about new technologies, such as bioengineered food, is the media. Media coverage of food technologies has a strong impact on how consumers perceive those technologies, as well as consumer attitudes and behaviors toward genetically edited food products [29]. Several studies have discussed the role of the media in reporting on genetically modified food in several countries around the world, such as Japan, the United States, and countries in the European Union (EU). It was found that, in general, the media tends to have a negative approach towards biotech based food products in general, and the media tends to act more as a risk communication entity rather than through science communication. Moreover, a paper by McCluskey in 2015 has argued that positive news coverage is often dominated by the negative coverage [30,31,32].

Therefore, and based on the articles discussed above, we have concluded that the general public tends to have concerns about genetically engineered food products. Further, we have concluded that knowledge and education play an important part in increasing the adoption rate of genetically modified food products. 

## 3. Regulations of Genetically Modified and Genetically Edited Foods

Regulations for defining and labeling genetically modified food are considered well developed compared to the regulations that govern genetically edited food, due to the history of GMO food commercialization that has occurred since 1994 [33]. Regulations for GMOs are continuously being developed in several regions around the world to cooperate with the latest health and safety standards as well as to match the labeling requests of consumers.

This section provides a summary of the regulations that govern genetically modified food in comparison with the ones that govern genetically edited food across three regions: Japan, Europe, and the United States.

In terms of genetically modified food labeling, the regulations differ by region. However, they can mainly be categorized as mandatory labeling or voluntary labeling. In the voluntary cases, authorities give the manufacturer the option to label the genetically modified food product, unless it has significant differences from the conventional one in composition or allergenic potential, at which point, the labeling becomes mandatory. Mandatory labeling can be further divided into two main categories, pan-labeling, and designated product labeling. In the pan-labeling category, the products must be labeled if they contain genetically modified materials that exceed a certain threshold level defined by the regulatory authority, or if the product has significant differences from the conventional one. For designated product labeling, specific products identified as genetically modified, or in certain categories, must be marked as such.

In terms of genetically edited food, the regulations are still in the development stage, where each country is developing suitable regulations domestically, as there are no global regulations in this regard [34]. However, regulatory authorities are divided into two main regulatory approaches. The first approach is to consider genetically edited food as genetically modified food products that must have specific safety regulations. The other approach is to consider genetically edited food products as conventional products since, in many cases, there is no substantial distinction between genetically edited products and products developed by conventional breeding.

### 3.1. Regulations in Japan

In terms of genetically modified food products, designated Japanese authorities have confirmed that any crops or processed food that contain certain genetically modified materials must be labeled. Regulations for processed food products specify that if one of the top three ingredients in terms of product weight ratio contains genetically modified material, or if the genetically modified ingredients comprise more than five percent of the total weight ratio of the product, it must be labeled as a genetically modified product. However, in cases of genetically modified materials that cannot be detected in oil or sauce products, mandatory labeling is not required.

As we discussed in the introduction, genetically edited food is fundamentally different from genetically modified food in terms of methodology. Therefore, the Japanese Ministry of Health, Labour, and Welfare (MHLW) began to discuss how to regulate the food created by genome editing technology in September 2018. On March 27, 2019, the MHLW released its policy related to genetically edited food product regulations after studying 691 comments received from the public on the subject. The MHLW’s specialized committee proposed that any food created by genome editing technology that contains transgenes has to go through the same safety review as specified in current regulations for genetically modified food. If the food does not contain any transgenes, it will not be handled the same as the food created by DNA recombination technology [35]. The entity responsible for labeling and consumer protection in Japan is the Consumer Affairs Agency (CAA). In late 2019, the CAA announced that food created by genome editing technology does not require specific labeling to mark it, unlike GMO food, which is subject to a specific labeling policy as well as safety inspection [36].

### 3.2. Regulations in the European Union

The regulations that govern genetically modified food are considered the strictest regulations globally in terms of labeling because, in Europe, all genetically modified food products must be labeled as per the decision of the European Commission in 1998. The regulations and policies governing genetically modified food were updated in November 2003 to include genetically modified food products with undetectable genetically modified DNA. The regulations also stated that conventional food products containing 0.9% or more genetically modified material must have genetically modified identification labels.

On July 25, 2018, the European Union Court of Justice issued press release number 111/18, which stated that genetically edited food would be considered a subset of genetically modified food, since it is still created by a gene alteration methodology. Subsequently, all food created by gene editing technology would need to go through all of the rules and regulations that govern genetically modified food in terms of labeling and safety inspection. This court ruling was in contrast to Japanese regulations that completely differentiate between genetically edited food and genetically modified food [37].

European research labs and biotechnology experts have considered this ruling as a setback for the innovation in the food technologies in Europe. They expect it to be a hindrance to scientific development in the region since many other countries around the world are willing to apply gene editing technology to help solve economic challenges related to food and agriculture.

### 3.3. Regulations in the United States

The regulation of genetically edited food in the United States is still in the development phase. However, several drafts and official meetings have been conducted in this regard. On January 11, 2017, the FDA issued a draft with the title “Genome Editing in New Plant Varieties Used for Food,” which was intended to be an essential guideline for regulation and gain the public opinion about the process [38]. The FDA has defined genetically edited food as the food created by gene editing technology that allows scientists to make specific changes in a specific site of the genome. Gene editing can be conducted using several methodologies, including CRISPR, Zinc Finger Nucleases (ZFNs), Transcription Activator-Like Effector Nucleases (TALENs), and Oligonucleotide Directed Mutagenesis (ODM) [39]. Furthermore, the FDA has regulated genetically edited food by requiring it to meet the same food safety requirements as food created by traditional breeding. For crops created by gene editing technology, the U.S. Department of Agriculture (USDA) plans to consider them as traditional breeding products since similar changes can happen using traditional breeding technology. However, the FDA is trying to take a different approach in regulating genetically edited livestock because it may be similar to pharmaceutical drug regulations [40].

### 3.4. Analysis of the Regulations

Regulations for genetically modified food differ fundamentally between Japan, Europe, and the U.S. The Japanese and European systems follow mandatory regulations, whereas the U.S. follows voluntary regulations, as explained above. Regulations for genetically modified food labeling tend to be stricter in the European Union since all products that contain genetically modified food ingredients must be labeled, even if the GMOs are undetectable in the end product. In addition, regulations for genetically edited food products follow the same system as genetically modified food in Europe, which has created a dilemma for genetically edited food labs in the region since the decision hinders innovation in the field of genetically edited food. Furthermore, the European decision to consider genetically edited food products as genetically modified food has created another problem for the food safety laboratory and inspection units, since inspectors are now required to test if the food contains genetically edited substances; however, scientists continue to struggle to find an accurate test because genetically edited food can be easily mistaken for food created by conventional breeding methods [41]. Table 1 shows the analysis of the regulations for both genetically edited and genetically modified food. It has been created based on a study from the Centre for Food Safety established by the Hong Kong government [42], along with analyses from other governmental sources, such as the United States Department of Agriculture (USDA), Food and Drug Administration (FDA), Ministry of Health Labour and Welfare (MHLW) in Japan, Consumer Affairs Agency (CAA) in Japan, and the European Commission (E.C.).

Based on the aforementioned analysis, we believe that the system in Japan opens the doors for more innovation in genetically edited food products. The gene editing field is an innovative approach to solve several challenges related to food safety and security, and it is fundamentally different from the gene modification approach.

Japan has several challenges in terms of agriculture, and genetically edited food may be a significant step toward solutions to tackle those challenges. The regulations governing genetically edited food in Japan are quite attractive for biotech companies to innovate in this field. The regulations in Japan allow companies to develop their products without any additional inspections compared to genetically modified food. However, the question remains if Japanese consumers are willing to consume genetically edited food products since they are a new type of product that also has the word “genetics” in their name. We have reviewed several scientific articles that show that consumers have associated the term with several health concerns. Therefore, this paper is trying to examine further how consumers will perceive genetically edited food in Japan and determine the factors that may enhance their level of willingness to purchase these products in the future. We believe that if we can identify those factors, it will help us understand consumer purchasing behavior and how we can implement a plan to enhance interest. These actions will give Japanese biotech companies and gene editing food researchers more incentive to innovate in this field by providing evidence of acceptance of this type of product among the public.

## 4. Research Framework

### 4.1. Questionnaire Design

The structural equation modeling (SEM) approach was utilized to measure the factors influencing the adoption rate of genetically edited food among youth in Japan. The factors subject to the study were as follows: Knowledge, Attitude Towards Technology, Perceived Benefits, Perceived Risks, Trust, and Willingness to Purchase. Every factor was measured by several questions, as stated in Appendix A.

The survey was designed to be conducted twice to measure the effect of science communication on enhancing the knowledge and adoption of genetically edited food. The process of the survey was to provide the questions digitally in an education session where the respondents answer the first survey. After the first survey, the respondents viewed a short presentation for five minutes that provided more knowledge about the genome editing technology. The respondents answered the same survey one more time to measure the difference in results before and after the intervention. The survey also measured several demographic factors, such as gender, nationality, hometown, and living status. To ensure validity and construct reliability of the research tool, the survey was conducted twice for continuous updates regarding clarity and question relevance.

The five-minute presentation, which was used as an intervention between the first and second surveys, provided brief information about the difference between natural and refined food as well as the main differences between organic food, non-GMO food, genetically modified food, and genetically edited food. The presentation also introduced the purpose of GMO food and the example of Bt-corn, which is widely used in the U.S. The presentation also showed how the scientific community and governments had deemed GMO food safe to consume; however, there are still increasing consumer concerns about GMO food as well as other risks from organic food or non-GMO food. Participants learned how genetically edited food differs from genetically modified food given that gene editing technology does not utilize any transgenes in the process and provides a more accurate, timely, and cost-efficient alternative for genetically modified food as introduced by scientific communities around the world. At the end of the presentation, the cases of the American company that introduced zero trans-fat soybean cooking oil, as well as the Japanese universities who developed gene-edited fish containing 20% more meat than average, were briefly introduced as examples of the impact of gene editing technology in food as shown in Appendix B.

### 4.2. Data Collection

Since the central aspect of this study is to explore the acceptance of gene editing technology for food among youth in Japan, the survey was conducted at Ritsumeikan University in Osaka, Japan. The survey was conducted on December 24, 2019, and attended by nearly 250 students with an average age of 20 years. The students’ educational backgrounds were in economics and business, which was intentional, to avoid respondents with science or biology backgrounds, to represent the general population, who do not have any previous scientific experience concerning genetically edited food or genetically modified food. The respondents participated in the survey voluntarily and without any financial incentive to avoid any data collection bias. A group of 216 respondents answered the first survey, and then attended the short presentation about genome editing technology, and 207 respondents retook the survey. After filtering the date and matching the ID numbers of the respondents for the first and second surveys, we received 180 completed responses.

The statistical survey results for the 180 completed samples show that 47% of respondents were female, while 53% were male. The majority of respondents were Japanese nationals (92%), while 6% were Korean nationals, and 2% were Chinese nationals. However, all of the respondents were residents of Japan. Moreover, 72% of respondents had an urban hometown, whereas 28% had a rural hometown. The majority of respondents said that they live with their family (71%), but 27% selected living alone, and 2% specified living with friends, as shown in Table 2.

## 5. Theoretical Model

There is a lack of research articles discussing the acceptance of genetically edited food using the SEM method due to the uniqueness of the genetically edited food approach and the fact that it is not commercialized on a broader level. Therefore, this research has utilized public attitudes toward biotechnology, identified by Pardo [43], as well as Fishbein’s multi-attribute model that explores the relationship between beliefs about a particular object and the attitude towards it [44]. In addition, several papers related to the acceptance of genetically modified food have been utilized in creating this model. The model measures the following different factors: Knowledge (KN), Attitude Towards Technology (ATT), Perceived Benefits (PB), Perceived Risk (PR), Trust (TR), and Willingness to Purchase (WTP) as presented in Figure 1.

### 5.1. Knowledge (KN)

In general, knowledge about a particular subject has a high impact on the attitude towards it. Although the public has a low level of understanding of genetically edited food, knowledge about biotechnology and its impact on modern society has increased dramatically due to advances in communication technologies and science communication in general. Gaskell has argued that knowledge plays an essential role in terms of perceived risk, following a study showing that limited knowledge regarding genetically modified food can increase the level of risk perception [45]. Therefore, our hypothesis (H) for the knowledge factor was set as follows:

**Hypothesis** **1.**
*Knowledge has a direct effect on Trust.*


**Hypothesis** **2.**
*Knowledge has a direct effect on Perceived Benefits.*


**Hypothesis** **3.**
*Knowledge has a direct effect on Perceived Risks.*


For this factor, we studied the level of knowledge that the respondents had in terms of genetically modified food and genetically edited food, as well as if they could differentiate between them. In addition, we determined if the respondents knew the potential for utilizing gene editing technology in food, animals, and human health, as well as their willingness to develop further knowledge about this technology.

### 5.2. Attitude Towards Technology (ATT)

Several scientific articles have argued that if a consumer trusts a certain technology, the general new technology adoption rate for the same consumer may increase [46,47,48]. In this construct, we argue that the consumers who have a high interest in adopting new technologies will also have an interest in understanding information about genetically edited food, and consequently, may have a higher interest in purchasing and consuming genetically edited food, since it is considered a technology-based product. Another study that aims to explore attitudes toward GMOs has found that the majority of respondents believe that science and technology are essential for the development of humankind and, specifically, for local development as well as the local economy. In addition, the study has found a positive correlation between attitudes toward technology in general and the belief that GMO food can increase agricultural production [49]. Therefore, the hypotheses for the Attitude Towards Technology factor were developed as follows:

**Hypothesis** **4.**
*Attitude Towards Technology has a direct effect on Knowledge.*


**Hypothesis** **5.**
*Attitude Towards Technology has a direct effect on Trust.*


### 5.3. Trust (TR)

Mossialos has argued that there is a knowledge gap in the area of genetics alteration, which is caused by the low efficiency of benefits and risks communication due to widespread sources discussing this matter and scientific uncertainty [50].

A study by Moon and Balasubramanian has shown that consumers in the United States and the United Kingdom consider the primary information source about genetics technology to be the government. Thus, trust in the government and scientists is considered an essential part of shaping public trust in genetics technology [51]. Another study was conducted in Colorado to identify the trusted sources of information about Agriculture, Biotechnology, and Food Information. It has conducted a survey to identify the most trusted source of information about food quality and safety among the residents of Colorado. The survey was conducted using four levels Likert scale survey, where one represents the most trusted and four represents the least trusted. It was found that universities and research organizations, as well as the Colorado Department of Agriculture, are the most trusted with a mean of 1.6, where social media was the least trusted source with a mean of 2.9 [52].

In this construct, we aimed to study the effect of having higher trust in the information sources or the key players in the gene editing field on adopting genetically edited food. Therefore, our hypothesis was as follows:

**Hypothesis** **6.**
*Trust has a direct effect on Willingness to Purchase genetically edited products.*


In this factor, we studied the level of trust that the respondents had toward the scientists working in the biotech industry, academic members, government information regarding genetically edited food, and companies utilizing gene editing technology, as well as in the sources providing information about gene editing technology. 

### 5.4. Perceived Benefits (PB)

Several previous studies have indicated that engagement with biotechnological products is positively correlated with perceived benefits [53,54,55]. Therefore, the following hypothesis was created:

**Hypothesis** **7.**
*Perceived benefits have a direct effect on Willingness to Purchase genetically edited products.*


In this factor, we investigated the level of consumer perceived benefits toward genetically edited food products by assessing several factors, such as the benefits to human health, benefits for animal health and comfort, positive effects on the local economy and agriculture, and the impact of solving hunger issues in developing countries.

### 5.5. Perceived Risks (PR)

A study by Bredahl has suggested that there is a connection between perceived risks and attitudes toward genetically modified food. The study has further determined that consumer acceptance of genetically modified food tends to be very low because the perceived risks are higher than the perceived benefits, putting further emphasis on measuring the difference between perceived benefits and risks to measure attitudes toward genetically modified products [56]. Therefore, our hypothesis was proposed as follows:

**Hypothesis** **8.**
*Perceived risks have a direct effect on Willingness to Purchase genetically edited products.*


### 5.6. Willingness to Purchase (WTP)

In this study, we examined several factors that influence consumers’ willingness to purchase genetically edited food products. We have tested how the consumers’ willingness to purchase genetically edited food products are directly affected by the trust in the genetically edited food industry stakeholders (H6), the benefits of genetically edited food perceived by consumers (H7), and the risks of genetically edited food perceived by consumers (H8). In this construct, we are also exploring different attribute factors that may encourage the consumer to purchase genetically edited food products, such as having the products at lower prices compared to the conventional ones, or containing more nutrients than the conventional ones.

## 6. Data Analysis

For SEM analysis, IBM SPSS Amos (v. 26) software was used to analyze the theoretical model responses for the first and second surveys as well as to test the hypotheses stated in the previous section. The SEM method was used since it can detect measurement errors as well as estimate the path coefficient of the proposed model in a visualized way. The SEM method has had a substantial impact on research related to social studies or studies that measure different factor effects [57].

### 6.1. Construct Reliability and Validity

Exploratory factor analysis (EFA) was conducted after the intervention to validate the reliability and validity of the constructs. For this purpose, we utilized IBM SPSS Statistics (v. 26) software with factor analysis functionality. To ensure the highest level of compatibility with IBM Amos, we employed the method of maximum likelihood for factor extraction since it is the same method utilized by Amos with a fixed number of five factors to be extracted and maximum iterations for convergence at 25. The analysis was conducted using Promax rotation with the Kaiser normalization method to identify the correlated factors. To ensure a high level of factor effects, we automatically suppressed small coefficients with an absolute value below 0.3. After conducting multiple rounds of EFA, several variables were deselected from the analysis, especially the ones that had a loading of less than 0.3 or were loaded in two different factors in the analysis, including KN7, TR3, TR6, TR7, WTP7, ATT3, ATT4, ATT5, and ATT7, to ensure a higher model fit. The finalized exploratory factor analysis contains 33 items instead of the 42 items from the original questionnaire, as shown in Appendix C.

We measured the Kaiser-Meyer-Olkain (KMO) value using SPSS to measure the sampling adequacy, and it was found to be 0.904. Since the resulting KMO value was between 0.9 and 1, the analysis is considered “marvelous,” according to Kaiser, which reflects the highest level of sample adequacy [58]. All items for all factors loaded more than 0.40. Therefore, all items were considered valid to be used [59].

The analysis covered the internal reliability for all of the constructs as well as external reliability by covering Cronbach’s alpha for all items tested. Cronbach’s alpha was confirmed to be sufficient for all constructs since it exceeded the recommended value of 0.7 [60]. Therefore, the reliability and validity of the model using the survey data sets collected before and after the intervention were established, as shown in Table 3. Model Construct Reliability.

### 6.2. Model and Hypothesis Testing

Since the model met all of the fitness, reliability, and validity requirements using both datasets of responses from before and after the intervention, path analysis for the model was conducted. The model tested the effect of knowledge on perceived benefits, trust, and perceived risks, the effect of attitude towards technology on knowledge and trust, and the effect on willingness to purchase resulting from perceived benefits, trust, and perceived risks. All of the hypotheses have been confirmed and accepted after measuring the P-values of all paths in the model of the survey after the intervention. All P-values were accepted with statistical significance below a value of 0.05. However, in the survey before the intervention, the paths of Attitude Towards Technology to Knowledge and Knowledge to Trust had P-values of 0.647 and 0.091, respectively, so we failed to accept these hypotheses as shown in Table 4 and Table 5.

### 6.3. Path Coefficient

The model passed statistical significance requirements by having a P-value less than 0.05 for all of the second survey. However, it failed in two paths on the first survey, namely Attitude Towards Technology to Knowledge and Knowledge to Trust. We found that all of the paths hold positive values for coefficients except the path of Perceived Risks to the Willingness to Purchase, which holds a negative coefficient and is compatible with the majority of studies conducted in this field given that perceived risks negatively affect willingness to purchase. The study also highlights the effect of knowledge and the differences in coefficients before and after the intervention. Before the intervention, the paths between Knowledge to Trust, Perceived Benefits, and Perceived Risks held the values of 0.11, 0.56, and 0.30, respectively. In path analysis of the second survey, we noticed that the values of the paths increased to 0.38, 0.77, and 0.36, respectively, which illustrates the effect of the Knowledge construct on Trust, Perceived Benefits, and Perceived Risks as shown in Figure 2 and Figure 3. In addition, the short presentation in the middle of the first and second surveys affected the Knowledge construct, and we found two significant differences between the respective models before and after the presentation in the critical path of Knowledge to Trust. The model fit has been measured to be adequate as shown in Table 6.

### 6.4. Willingness to Adop

To further verify the results, at the end of both surveys, we asked the respondents one direct question about whether they were willing to purchase genetically edited food. The answer options were presented as yes or no. We found a change in the final opinions of respondents; in the first survey, only 24% showed interest in purchasing genetically edited food products, but after the second survey, the number significantly increased to 41%. This result illustrates how science communication and increasing the knowledge of consumers through technology can increase their willingness to purchase, as shown in Table 7.

To further verify the statistical significance of the results mentioned above, a chi-square test was performed using SPSS. The asymptotic significance has been calculated to be less than 0.001, which indicates a high level of statistical significance for the differences shown in Table 8.

## 7. Study Limitations

In this study, we are tackling three main themes, namely; the regulations comparison, genetically edited food knowledge, and the structural equation modeling analysis for identifying the factors that can affect the willingness to purchase the genetically edited food products. The study has three main limitations, which will require further research work to tackle them.

The first limitation in the regulatory comparison part. It is clear that the framework and regulations for genetically edited food are still in the development stage in several areas around the world, and it is not yet solid or well established in comparison with the genetically modified food regulations. The comparison in Europe and the United States are in the development stage, and we encourage future research work to compare the final version of the regulations that govern genetically edited food once released or approved.

The second limitation is represented in the literature review for the willingness to adopt genetically edited food or the factors that can influence it. Considering that this topic is quite new, and there is limited knowledge of the genetically edited food products among consumers, the number of literature that tackles this part is considered to be very limited in comparison with the genetically modified food. Therefore, most of the literature that we have reviewed were mainly tackling consumer perception of genetically modified food. Therefore, we highly encourage more researchers to conduct more surveys and experiments to measure consumer perception of genetically edited food in the future.

The third and main concern while conducting this study is that the survey was distributed among Ritsumeikan University students in Japan who happened to be in the same age bracket, as well as from similar social backgrounds. Therefore, the results of this survey should not be interpreted as a reflection for Japan nationwide due to the tight demographic and geographic proximity of the survey respondents. Thus, we highly recommend further research efforts to validate the results by applying this model to samples of different ages and from different cities and/or social backgrounds to verify the degree to which this model represents the overall population of youth in Japan.

Although we believe that the study has several limitations, the study can be considered as a start for more researches in this field. We are planning to conduct more related and complementary studies in the future, and we highly welcome the collaboration with more researchers interested in the field for future and more inclusive studies.

## 8. Conclusions

Biotechnology is one of the most advanced scientific fields with constant innovation that can lead to impressive results. However, the field is also highly regulated due to its critical nature. The regulations are often associated with being anti-innovative by scientists in this field. In the genetically edited food field, regulations around the world are still immature and ambiguous, especially in terms of labeling. Therefore, we consider this paper as a start for comparing regulation approaches around the world, and we aim to conduct more comparisons once there are concrete regulations set by authorities in different countries.

Genetically edited food is considered a new term. Thus, it is expected that the general population will have a lack of understanding of the differences between genetically modified food and genetically edited food. Therefore, we have created an intervention featuring a short educational presentation that highlights the aforementioned differences. We have observed significant changes in the results before and after the intervention, which indicates the impactful role that science communication plays in this regard. As a result, we highly encourage more researchers to investigate the role of science communication in increasing the adoption of biotechnological products and also collaborate to create easy-to-understand materials for the general public using basic terms. The responsibility and importance of science communication are not less critical than scientific achievements. Therefore, we advise that more researchers adopt this approach to help the public to understand scientific achievements and the innovation behind them, since current media sources may not accurately and effectively communicate these topics to a wider audience.

Based on this study’s results, we advise considering science communication is the main factor that can affect willingness to purchase and the success of genetically edited food products in the market. Based on the survey data, we have also noticed that consumers trust the information and regulations of governments. Therefore, we advise governments to investigate further the feasibility of establishing online portals that provide education about genetically edited food in simple terms for general consumers. We also advise universities and research labs to investigate the possibility of creating a joint communication platform that delivers the latest results of research on the benefits or risks of genetically edited food products to the public. Other potential outreach approaches include developing cooking shows and YouTube videos, which can briefly introduce genetically edited food products and utilize them in an educational cooking session. Researchers in food-related technologies can also introduce the concept of genetically edited food to researchers in different fields via gathering opportunities such as Science Café, which is organized by the Japan Society for Bioscience, Biotechnology and Agro-chemistry, or any other similar events that aim to increase awareness of different scientific concepts.

Based on current regulations in which Japan shows a direction of being science-based, we believe that Japan would be an attractive place for biotech companies to expand in the genetically edited food field. Therefore, we advise investigating further the efficiency of establishing investment schemes and incentives that attract leading biotech companies and innovative international start-ups to move their operations to Japan in order to gain a higher share of the Japanese market, as well as add to Japan’s domestic economy and further develop Japan into a hub of innovation in food technology-related fields.

## Figures and Tables

**Figure 1 ijerph-17-02935-f001:**
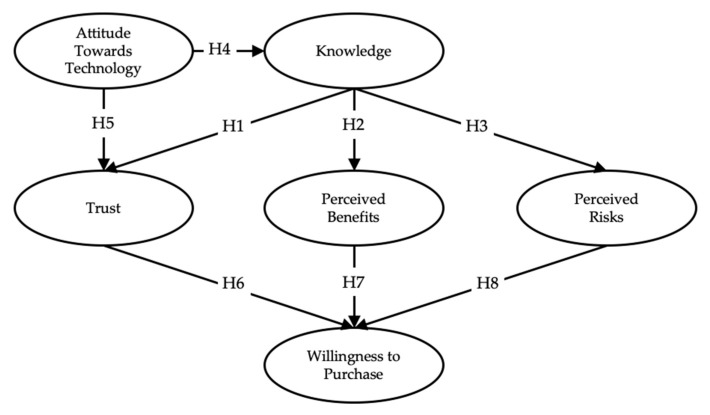
Model to be tested for hypotheses (H).

**Figure 2 ijerph-17-02935-f002:**
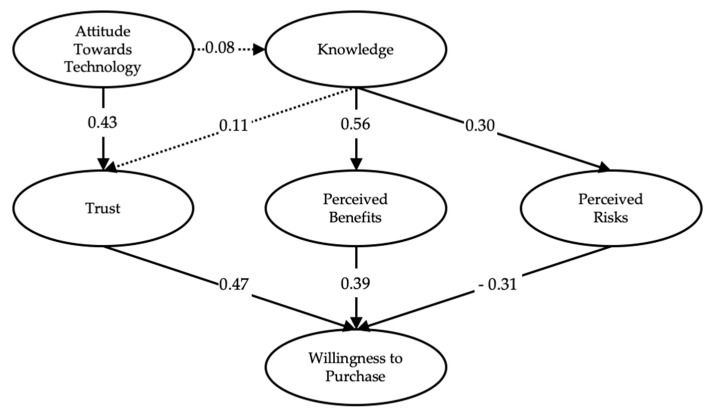
Standardized estimates for the edited model (First Survey).

**Figure 3 ijerph-17-02935-f003:**
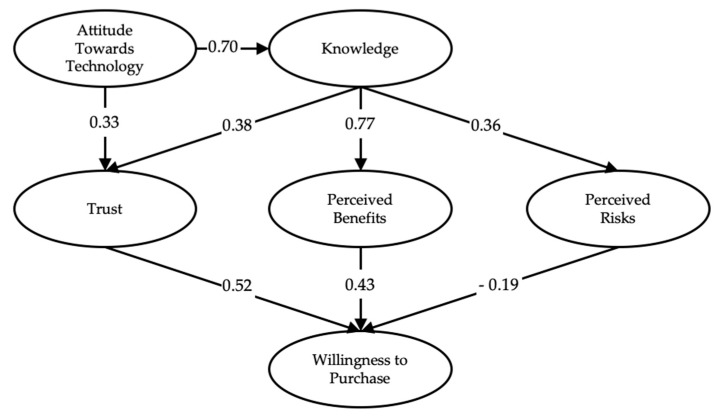
Standardized estimates for the edited model (Second Survey).

**Table 1 ijerph-17-02935-t001:** Genetically modified food (GMO) and Gene Editing Regulations in Japan, Europe, and the USA (Reviewed: March 3, 2020).

Comparison Aspect	Japan	Europe	USA
GM Regulation Type	Designated products labeling	Pan-Labeling	Voluntarily Labeling
GM Regulation Advantages	Provides moderate notification for consumers in case the food components contain GM ingredients higher than a certain level	Provide clear identification for consumers regarding the GM components	Provides explicit notification for customers with a specific allergy or dietary need
GM Regulation Disadvantages	Does not meet the needs of the consumers who would like to ensure that the food does not contain any GM materials at all	Difficult to ensure its full enforcement by the government due to the limitation of the dedication for GM components in many cases	Does not provide clear identification for GM products for the customers sensitive to this matter
Gene-edited food regulation	Yes	No	In the development stage
Gene-edited food commercialization	No	No	Yes (Soy Oil)
How gene-edited food categorized	As conventional food	As genetically modified food	In the development stage; however, USDA has a potential approach of having gene-edited food as conventional, and animal products as genetically modified
Gene-edited food requires safety testing	No, a voluntary notification system may apply	Yes, require full testing as genetically modified food	In the development stage.
Gene-edited food requires specific labeling	No	Yes, similar to GMO	In the development stage.

GM: Genetically Modified, MHLW: Ministry of Health, Labour and Welfare of Japan, CAA: Consumer Affairs Agency; E.C.: European Commission, USDA: United States Department of Agriculture, FDA: Food and Drug Administration.

**Table 2 ijerph-17-02935-t002:** Descriptive statistics for participants’ demographics.

Characteristics	Variables	Number of Participants	Percentage of Participants (%)
Gender	Female	84	47%
	Male	96	53%
Nationality	China	3	2%
	Japan	166	92%
	Korea	11	6%
Hometown	Rural	50	28%
	Urban	130	72%
Living with	Alone	49	27%
	Family	128	71%
	Friends	3	2%
Total		180	100%

**Table 3 ijerph-17-02935-t003:** Model Construct Reliability.

Construct	Cronbach’s Alpha	Mean	Variance	Std. Deviation	N of Items
*KN*	0.902	*20.31*	*19.959*	*4.467*	*6*
*PB*	0.870	*23.99*	*22.760*	*4.771*	*7*
*PR*	0.877	*23.11*	*19.574*	*4.424*	*7*
*TR*	0.823	*13.29*	*7.771*	*2.788*	*4*
*WTP*	0.933	*19.61*	*24.898*	*4.990*	*6*
*ATT*	0.702	*7.45*	*2.092*	*1.447*	*2*

KN: Knowledge, PB: Perceived Benefits, PR: Perceived Risks, TR: Trust, WTP: Willingness to Purchase, ATT: Attitude Towards Technology.

**Table 4 ijerph-17-02935-t004:** Estimated regression weights of the structural model (First Survey)

Factors	Estimate	S.E.	C.R.	P	Result
KN🡸ATT	0.084	0.115	0.872	0.383 *	Not Supported
PR🡸KN	0.301	0.073	3.288	0.001	Supported
TR🡸KN	0.114	0.063	1.196	0.232 *	Not Supported
PB🡸KN	0.563	0.086	4.829	0.000	Supported
TR🡸ATT	0.438	0.096	3.601	0.000	Supported
WTP🡸PR	−0.319	0.095	−3.818	0.000	Supported
WTP🡸TR	0.479	0.145	4.555	0.000	Supported
WTP🡸PB	0.399	0.114	4.287	0.000	Supported

***** indicates P-Value higher than 0.05 (not significant); KN: Knowledge, PB: Perceived Benefits, PR: Perceived Risks, TR: Trust; WTP: Willingness to Purchase, ATT: Attitude Towards Technology.

**Table 5 ijerph-17-02935-t005:** Estimated regression weights of the structural model (Second Survey)

Factors	Estimate	S.E.	C.R.	P	Result
KN🡸ATT	0.701	0.128	6.144	0.000	Supported
PR🡸KN	0.369	0.076	4.06	0.000	Supported
TR🡸KN	0.388	0.119	2.907	0.004	Supported
PB🡸KN	0.773	0.101	7.632	0.000	Supported
TR🡸ATT	0.331	0.143	2.319	0.02	Supported
WTP🡸PR	−0.194	0.085	−3.042	0.002	Supported
WTP🡸TR	0.527	0.108	6.141	0.000	Supported
WTP🡸PB	0.437	0.095	5.134	0.000	Supported

KN: Knowledge, PB: Perceived Benefits, PR: Perceived Risks, TR: Trust; WTP: Willingness to Purchase, ATT: Attitude Towards Technology.

**Table 6 ijerph-17-02935-t006:** Model Fit Index.

Construct	Cronbach’s Alpha	Value
Χ^2^	Model Chi-Square	673.537
df	Degrees of Freedom	411
Χ^2^/df	Chi-Square/Degrees of Freedom	1.639
CFI	Comparative Fit Index	0.926
RMSEA	Root Mean Square Error of Approximation	0.060

**Table 7 ijerph-17-02935-t007:** Willingness to purchase (N. 180).

Survey	Yes		No	
	N.	%	N.	%
First	43	24%	137	76%
Second	73	41%	107	59%

**Table 8 ijerph-17-02935-t008:** Chi-Square test for Willingness to purchase results (N. 180).

	Value	df	Asymptotic Significance (2-sided)
Pearson Chi-Square	70.358 ^a^	1	0.000
Number of Valid Cases	180		

a. The minimum expected count is 17.44.

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
