# Peer review of "Exploring Factors Affecting the Acceptance of Genetically Edited Food Among Youth in Japan"

_ijerph, 2020, doi:10.3390/ijerph17082935_

Round 1

Reviewer 1 Report

The paper addresses an important topic. The paper requires a substantial up-date of the literature. The part on the approval and labeling includes a number of flaws that need to be corrected. The empirical part of the paper looks fine from the technical point of view but the results need to be embedded in the current literature. Otherwise they remain meaningless. More comments attached in the pdf.

Author Response

  • The paper addresses an important topic. The paper requires a substantial update of the literature. The part on the approval and labeling includes a number of flaws that need to be corrected. The empirical part of the paper looks fine from the technical point of view but the results need to be embedded in the current literature. Otherwise they remain meaningless. More comments attached in the pdf.
    • Thank you very much for your constructive comments, we completely agree with your point of view. Accordingly, we have completely updated the literature review section (section 2), it now has now nearly 20 references. We have reviewed also more updated articles as advised. In the literature review part, we have added more related literature to the results of the study. We have focused on how the public perceive the GMO products as a risk, especially in Japan, United Sates, and Europe. Also, we have added several papers that shows the positive effect of education and knowledge in increasing the adoption of bioengineered food products, which is highly connected and aligned with the study results, as we have shown by our study survey that increasing the knowledge via science communication, has positive effect on the willingness to purchase genetically edited food products in the future. Regarding the regulations part, we have reviewed also several papers and reports which we have stated in the literature review part, as well as the introduction part. We have added more updated literature from the year 2015 to 2019.
  • The term GMO has a legal definition in the EU but not in the US. You need to make this clear from the beginning. This has implications for regulations. Details can be found e.g. in Dries et al. 2019. EU Bioeconomy Economics and Policies, Volume II.
    • Thank you very much for pointing out this part, we acknowledge that there is no federal law that govern GMO yet, and there is no concrete legal definition for it in the United States. Accordingly, and based on your kind guidance, we have added a complete paragraph that explains the framework in the United States that consists of the USDA, EPA and FDA. Also, we have added the definitions stated by the European Union explaining the GMO, which was stated in the EU directive, as we have pointed out in second paragraph of the introduction section, page two, lines 54-65 (using simple markup setting in Microsoft Word teach changes function).

  • Add a source, e.g. Eriksson et al. (2019) in New Phytologist.
    • Thank you very much for introducing us to such an important and updated source. Accordingly, we have added the recommended source, which gave us more clear vision regarding the regulatory approach of the genetically edited products. The recommended source has been added as source no. 11, in line 72.

  • See comment above. You need to clear with the terminology being used.
    • Thank you very much for your comments regarding the terminology of the GMO. We acknowledge that there is no unified terminology for the GMO, however there are several definitions stated by the WHO, EU, FDA and USDA. We have identified the different definitions as you have kindly advised, and it was added in the first paragraph of section 1.1. “Genetically Modified Food”, also the sources for the definitions have been added as source no. 12 and 13 in the lines 88 and 90 respectively.

  • Moderate English changes required 
    • Thank you very much for your comment, we acknowledge that there are opportunities to enhance the English writing style. Due to the fact that none of the co-authors is native English speaker, the manuscript was professionally reviewed by specialized academic proofreading entity, also we have conducted a second round of proofreading using a specialized academic writing proofreading platform. We have conducted several grammatical and punctuation edited in the whole document.

Reviewer 2 Report

General comments

- The literature review section is woefully inadequate. There is a substantial amount of literature on consumer perceptions that hasn’t been reviewed. The research by Jason Lusk, Brandon McFadden, Lynn Frewer, Jill McCluskey and others is all missing. This section needs to be significantly increased to merit a full review of the literature.

- Pages 9 list the 6 factors used in the model, yet only 5 are discussed. Subsection 5.3 discusses trust, yet the hypothesis is about willingness to purchase. It appears that text is missing as there is no subsection on willingness to purchase, nor a hypothesis on trust.

- Section 7 provides little value to the paper, with the exception of the last 2 paragraphs. Suggest deleting this section and moving these 2 paragraphs to the Discussion section.

- Given the limited number of participants, tight demographic and geographic proximity, a caveat needs to be included in the paper that highlights the limitations of the study. Readers need to be aware that this should not be interpreted as a national sample.

- My main concern with the paper is that the recommendations made in the discussion section lack the demographic confidence to make them. The study is of 180 students at one university, which is not rigourous enough to make the recommendations you have provided. The discussion section needs to be revised so that it is more of an advisory discussion, not a recommendation discussion.

Specific comments

- Abstract, lines 17-18, you state that Japan’s regulations are the most accommodating anywhere. GM crops have not been approved for production in Japan, so this statement isn’t an accurate one. Regulatory approval times for GM food products for import are very tedious and lengthy. The approval time for the decision to import GM papaya was 12 years. This is not the definition of “most accommodating”. This sentence needs to be removed or substantially revised to reflect the accuracy of the Japanese regulatory system delays. https://apps.fas.usda.gov/newgainapi/api/report/downloadreportbyfilename?filename=Japan%20approved%20GM%20papaya_Tokyo_Japan_12-19-2011.pdf

- P. 2, lines 52-52, you say that GM crops are “used all over the world”. This is not an accurate statement. GM crops are grown in 26 of the world’s 195 countries. Revise for accuracy. http://www.isaaa.org/resources/publications/briefs/54/executivesummary/default.asp

- P. 2, line 72, you say that GMO is the alteration of the DNA of plants for use in food. There are GM yeasts and bacteria that are common in the baking and dairy industries, such as in the production of cheese. This could be revised to more accurately reflect the food industry’s wide adoption of GM technology.

- P. 2, line 85, why do you only discuss the literature from 2002-2011? The National Academies of Science released a substantial review in 2016. The EU has released reports in the past several years as well. This should be revised to more accurately reflect publications post 2011.

- P. 2, line 87, “safety hazard” is an oxymoron, you can’t have both. Delete ‘safety’.

- P. 3, line 98, inserting a small number of foreign genes into a plant through genetic modification is not a “radical change” given that plants have genomes containing millions of genes.

- Figure 1 provides no useful information and can be deleted.

- P. 3, line 155, there is no global regulatory capability for gene edited crops or foods. All crop and food regulations are at the domestic level. See H.-G. Dederer and D. Hamburger (ed.), Regulation of Genome Editing in Plant Biotechnology: A Comparative Analysis of Regulatory Frameworks of Selected Countries and the EU, for a comparison of various national regulatory perspectives.

- P. 6, lines 230, the acronyms CAA and EC have not been previously used, spell in full.

- P. 6, lines 235-239, what methodology allows you to ascertain “the best approach for supporting gene edited food innovation”? If the Japanese regulatory system approves gene edited food as safe, implementing a labeling system undermines the regulatory competence of regulatory agencies. To be able to make this statement, you need to provide references for why this is the best approach. If not able to, this requires revision.

- Table 1, what GM products are available in Europe? Mandatory labelling has driven all GM products from the market. Some reference as to what GM labelled products are available for European consumers needs to be provided. The table also positively indicates that Europe has gene edited food regulation. The EU has GMO food regulations but nothing for gene editing, hence the CJEU ruling. The European Commission has funded a study to examine how to revise the existing regulatory framework, but as of now, nothing is in place to regulate gene edited foods.

- P. 8, line 285, Appendix C is the first appendix referenced, why is it not Appendix A?

- P. 9, line 316, Figure 1 shows nothing about the different factors.

- P. 10, line 335, you say several studies, yet only provide one reference. Add the additional references.

- P. 10, line 352, the Moon and Balasubramanian study is from 2001 and I have serious reservations that their results could be relevant for today’s public attitudes. This requires additional references to support the statement from the past 5 years of literature.

- P. 10, line 364, again, you state several studies yet only provide one reference. Add the additional references you are referring to.  

- P. 14, Table 6, the data is presented in a very confusing manner. Revise so that both surveys are on lines and yes and no are the column headers.

- P. 15, lines 470-471, the US has approved several gene edited crops without requiring any regulatory oversight, mushrooms and soy, to name but 2. I question the validity of the statement that Japan has the most accommodating regulatory system, especially in light of the fact that Japan doesn’t allow the production of any GM or gene edited crops. Unless you’re able to provide a metric for this assessment, it needs to be revised.

- P. 15, lines 477-478, how can you reach this conclusion when the survey had no connection to the question of biotech companies investing in gene editing? This sentence, is at best, a hypothesis.

- P. 15, lines 485-510, this has all been previously stated and is repetitive. Delete.

Author Response

  • The literature review section is woefully inadequate. There is a substantial amount of literature on consumer perceptions that hasn’t been reviewed. The research by Jason Lusk, Brandon McFadden, Lynn Frewer, Jill McCluskey and others is all missing. This section needs to be significantly increased to merit a full review of the literature.
    • Thank you very much for your constructive comments, we deeply appreciate your time and efforts spent in reviewing this part. Regarding the literature review part, we have made a significant review for it, and we have added the more updated literatures citing Ms. McCluskey and others. The literature review has focused more on how the public around the world perceive the genetically modified food. Also, we have focused on how knowledge and education play an important role in enhancing the adoption level of the genetically modified food products. The literature review is aligned with our finding for this study, as we have focused on the positive effect of knowledge and science communication in enhancing the adoption level of genetically edited food.

  • Pages 9 list the 6 factors used in the model, yet only 5 are discussed. Subsection 5.3 discusses trust, yet the hypothesis is about willingness to purchase. It appears that text is missing as there is no subsection on willingness to purchase, nor a hypothesis on trust.
    • Thank you very much for your comment. Regarding the section 5.3 discussing trust, there is a hypothesis that investigate the potential effect of trust on the willingness to purchase, namely H6. We acknowledge the comment regarding the willingness to purchase, accordingly and based on your kind advise, we have added subjection 5.6 that discusses the willingness to purchase, however, the hypothesis in this construct remain the same as it is connected with H6 about trust, H7 about perceived benefits, and H8 about perceived risks. Also, the literature review related to this construct remain the same as it is highly connected with the literature in subsection 5.3, 5.4, and 5.5 respectively.

  • Section 7 provides little value to the paper, with the exception of the last 2 paragraphs. Suggest deleting this section and moving these 2 paragraphs to the Discussion section.
    • Thank you very much for your constructive comments. We totally agree with your point of view, and we feel that it will make the content more focus. Accordingly, we have completely deleted the section 7 that summarizes the paper, except of the last two paragraphs as advised. The last two paragraphs have been transferred to the discussion section.

  • Given the limited number of participants, tight demographic and geographic proximity, a caveat needs to be included in the paper that highlights the limitations of the study. Readers need to be aware that this should not be interpreted as a national sample.
    • Thank you very much for your constructive comments. We totally agree with this point of view that the number of participants are very limited and can’t represent the national population. We have addressed this matter by creating a detailed limitation section number by section 8, to discuss all the limitations that we have faced. We have added a clear caveat about the study limitation and how the small number of participants can't represent the whole population. Also, we added another limitation which is the limited number of relative researchers that study the consumers behavior towards the genetically edited food.
  • My main concern with the paper is that the recommendations made in the discussion section lack the demographic confidence to make them. The study is of 180 students at one university, which is not rigourous enough to make the recommendations you have provided. The discussion section needs to be revised so that it is more of an advisory discussion, not a recommendation discussion.
    • Thank you very much for the comment. We acknowledge that the number of respondents is low and can’t represent national recommendation. As advised the whole discussion paragraph was rephrased to have more advisory tone rather than solid recommendation.

  • Abstract, lines 17-18, you state that Japan’s regulations are the most accommodating anywhere. GM crops have not been approved for production in Japan, so this statement isn’t an accurate one. Regulatory approval times for GM food products for import are very tedious and lengthy. The approval time for the decision to import GM papaya was 12 years. This is not the definition of “most accommodating”. This sentence needs to be removed or substantially revised to reflect the accuracy of the Japanese regulatory system delays.
    • Thank you very much for your comment. We totally acknowledge that part, that the Japanese regulation and approval time for GMO are lengthy. We have completely rephrased the part to focus on the regulation for the genetically edited food. The sentence was rephrased to “We found that the genetically edited food regulations in Japan are the most science-based, in the meaning that genetically edited food products are allowed to be sold without any safety evaluation”.

  • 2, lines 52-52, you say that GM crops are “used all over the world”. This is not an accurate statement. GM crops are grown in 26 of the world’s 195 countries. Revise for accuracy. http://www.isaaa.org/resources/publications/briefs/54/executivesummary/default.asp
    • Thank you very much for your comment, also for introducing the reference from ISAAA. After thoughtful study, we have modified the sentence to “ 70 countries around the world” for more accuracy as advised.

  • 2, line 72, you say that GMO is the alteration of the DNA of plants for use in food. There are GM yeasts and bacteria that are common in the baking and dairy industries, such as in the production of cheese. This could be revised to more accurately reflect the food industry’s wide adoption of GM technology.
    • Thank you very much for your comment, we have revised the definition of the genetically modified food completely in the first paragraph of the section 1.1., and we have added more wide definition as you have kindly advised. We have also pointed out the definitions created by the WHO, EU, FDA and USDA.

  • 2, line 85, why do you only discuss the literature from 2002-2011? The National Academies of Science released a substantial review in 2016. The EU has released reports in the past several years as well. This should be revised to more accurately reflect publications post 2011.
    • Thank you very much for your comment. We acknowledge the shortage of literature, accordingly, we have added more literature in this section from the past five years, especially the report released by the National Academies of Science, Engineering and Medicine, which was added as reference no.17, as kindly advised.

  • 2, line 87, “safety hazard” is an oxymoron, you can’t have both. Delete ‘safety’..
    • Thank you very much for your comment, we have deleted the word “safety” as advised for more focused text.

  • 3, line 98, inserting a small number of foreign genes into a plant through genetic modification is not a “radical change” given that plants have genomes containing millions of genes.
    • Thank you very much for your comment. We have rephrased the sentence by removing the word radically, for more accuracy description as advised.

  • Figure 1 provides no useful information and can be deleted.
    • Thank you very much for your comment, we acknowledge that the figure doesn’t add high value for the paper, and it was used for visualization purpose. Therefore, and based on your kind advise, the figure was deleted from the manuscript.

  • 3, line 155, there is no global regulatory capability for gene edited crops or foods. All crop and food regulations are at the domestic level. See H.-G. Dederer and D. Hamburger (ed.), Regulation of Genome Editing in Plant Biotechnology: A Comparative Analysis of Regulatory Frameworks of Selected Countries and the EU, for a comparison of various national regulatory perspectives.
    • Thank you very much for your comments, we acknowledge that there is no global regulatory in this regard. The original text meant to say that the efforts are being spent in several countries around the world (individually). Accordingly, we have rephrased this part to represent more accurate description as advised. The new sentence states that “In terms of genetically edited food, the regulations are still in the development stage, where each country is developing a suitable regulation domestically as there are no global regulations in this regard”, also we have added the recommended reference as advised and numbered by reference 35.

  • 6, lines 230, the acronyms CAA and EC have not been previously used, spell in full.
    • Thank you very much for your constructive feedback. We have added the full description for the acronyms as advised. We have listed the full acronyms in the lines of 304 to 306.

  • 6, lines 235-239, what methodology allows you to ascertain “the best approach for supporting gene edited food innovation”? If the Japanese regulatory system approves gene edited food as safe, implementing a labeling system undermines the regulatory competence of regulatory agencies. To be able to make this statement, you need to provide references for why this is the best approach. If not able to, this requires revision.
    • Thank you very much for your comment. Regarding this part, the sentence written was our own analysis and reading for the situation based on our previous experience comparing the functional food products regulations in Japan and around the World. Since that the core of the topic is different, therefore, we have deleted this part as advised from our manuscript as we do not have enough relative references to support this argument in the context of the genetically edited food.

  • Table 1, what GM products are available in Europe? Mandatory labelling has driven all GM products from the market. Some reference as to what GM labelled products are available for European consumers needs to be provided. The table also positively indicates that Europe has gene edited food regulation. The EU has GMO food regulations but nothing for gene editing, hence the CJEU ruling. The European Commission has funded a study to examine how to revise the existing regulatory framework, but as of now, nothing is in place to regulate gene edited foods.
    • Thank you very much for your comments. We acknowledge all the parts of the comment. We have deleted this part from the comparison table to avoid confusion for the readers. The GMO status has different aspects; cultivation, importation, production. In this section, we were influenced by the situation of MON810 cultivation especially in Spain. As advised, we have deleted the row from the comparison table, since that it speaks about GMOs, and does not affect the core of the study which is the genetically edited food acceptance. For the regulations of the genetically edited food in Europe, we acknowledge that there is no specific legal framework that govern the genetically edited food in Europe after the court of justice ruling as you have kindly advised. We meant in this section to mention that there is a court ruling at the moment that treat all the genetically edited food products as subsection of the genetically modified food. After further study, we have changed the status of regulation from “Yes”, to “No” as you have kindly advised.

  • 8, line 285, Appendix C is the first appendix referenced, why is it not Appendix A?
    • Thank you very much for your comments. We have reviewed all the appendixes through the manuscript and we have rearranged it and rename it based on the order referenced in the manuscript as advised.

  • 9, line 316, Figure 1 shows nothing about the different factors.
    • Thank you very much for your comment, we acknowledge the miss reference of the figure 1, and we have referenced figure 1 to represent the theoretical model of the study.

  • 10, line 335, you say several studies, yet only provide one reference. Add the additional references.
    • Thank you very much for your kind comments, we have reviewed and referenced two more studies in the same context to be 3 studies in total discussing how trust in certain technology can affect its adoption rate. The two new references are listed in the second line of subsection 5.2, line 403.
  • 10, line 352, the Moon and Balasubramanian study is from 2001 and I have serious reservations that their results could be relevant for today’s public attitudes. This requires additional references to support the statement from the past 5 years of literature.
    • Thank you very much for your comments, we acknowledge that part, and we have researched further the related paper in the last five years. We have referenced a study conducted in 2016 in Colorado, USA. The study showed that the residence of Colorado consider the Colorado department of agriculture and the universities, the most trusted source for biotechnology and food safety, where the social media is considered the least trusted. The paper was referenced by reference number 53, and was listed in line no. 428.

  • 10, line 364, again, you state several studies yet only provide one reference. Add the additional references you are referring to.
    • Thank you very much for your constructive feedback. We acknowledge the lack of the literature in this part as you have kindly mentioned. Accordingly, we have studied further papers and added two more references in this context. The new references added by the numbers of 55 and 56, in line 439.

  • - P. 14, Table 6, the data is presented in a very confusing manner. Revise so that both surveys are on lines and yes and no are the column headers.
    • Thank you very much for your constructive feedback, we acknowledge that part, and we have completely redesigned the table as advised.

  • - P. 15, lines 470-471, the US has approved several gene edited crops without requiring any regulatory oversight, mushrooms and soy, to name but 2. I question the validity of the statement that Japan has the most accommodating regulatory system, especially in light of the fact that Japan doesn’t allow the production of any GM or gene edited crops. Unless you’re able to provide a metric for this assessment, it needs to be revised.
    • Done, the sentence was deleted among other sentences in summary section.

  • - P. 15, lines 477-478, how can you reach this conclusion when the survey had no connection to the question of biotech companies investing in gene editing? This sentence, is at best, a hypothesis.
    • Thank you very much for your comment. We acknowledge this part, and we share the same view. We have rephrased this paragraph and deleted this part. Initially this sentence was intended to be our own analysis for the situation, that having no specific safety requirements to commercialize genetically edited product, therefore, we are expecting that the biotech companies can find Japan a good environment for growth in the field of genetically edited food. However, since that the sentence is quite broad and can cause misunderstanding for the sitation, the sentence was deleted as advised.

  • - P. 15, lines 485-510, this has all been previously stated and is repetitive. Delete.
    • Thank you very much for your constructive comment, we totally agree that deleting this part can make the text more focused. We have deleted the summary part completely except of two paragraphs that we added to the discussion section.

Round 2

Reviewer 1 Report

The authors have addressed the comments to my satisfaction.

Reviewer 2 Report

The authors have done a good job of revising the paper, which can now proceed to publication.